# Beta-Cell Mass in Obesity and Type 2 Diabetes, and Its Relation to Pancreas Fat: A Mini-Review

**DOI:** 10.3390/nu12123846

**Published:** 2020-12-16

**Authors:** Jun Inaishi, Yoshifumi Saisho

**Affiliations:** Department of Internal Medicine, School of Medicine, Keio University, Tokyo 160-8582, Japan; jinaishi@keio.jp

**Keywords:** beta-cell mass, diabetes, obesity, pancreas fat, human pancreas

## Abstract

Type 2 diabetes (T2DM) is characterized by insulin resistance and beta-cell dysfunction. Although insulin resistance is assumed to be a main pathophysiological feature of the development of T2DM, recent studies have revealed that a deficit of functional beta-cell mass is an essential factor for the pathophysiology of T2DM. Pancreatic fat contents increase with obesity and are suggested to cause beta-cell dysfunction. Since the beta-cell dysfunction induced by obesity or progressive decline with disease duration results in a worsening glycemic control, and treatment failure, preserving beta-cell mass is an important treatment strategy for T2DM. In this mini-review, we summarize the current knowledge on beta-cell mass, beta-cell function, and pancreas fat in obesity and T2DM, and we discuss treatment strategies for T2DM in relation to beta-cell preservation.

## 1. Introduction

The number of patients with diabetes mellitus continues to increase all over the world. There were 425 million patients with diabetes mellitus in 2017, and this number is expected to increase to 629 million in 2045 [1]. Since diabetes mellitus is associated with microvascular and macrovascular complications [2,3,4] and increased health care costs, optimal strategies to counter this pandemic are needed as soon as possible. More than 90% of individuals with diabetes mellitus have type 2 diabetes (T2DM). Both insulin resistance and beta-cell dysfunction are related to the development of T2DM. The increase of insulin resistance by obesity is a major risk factor for T2DM [5], and pancreatic fat contents increase proportionally with obesity. It has been reported that beta-cell dysfunction and apoptosis are induced by excess lipid accumulation in rodent islets [6,7]. In this review, we outline the current knowledge regarding beta-cell dysfunction and pancreas fat in obesity and T2DM, and discuss the treatment strategies for T2DM.

### Search Strategy

We used PubMed for this mini-review. The search terms were as follows: “beta-cell mass”, “pancreas fat”, and “type 2 diabetes”. We only considered articles written in English and published after the year 2000. Relevant articles were also collected from the personal databases of the authors.

## 2. Beta Cell Mass in Obesity and Type 2 Diabetes

Studies on human beta cells are important for establishing treatment approaches. However, the pancreas samples obtained surgically, at organ donation, or at autopsy are limited in number and each method has its own limitations. For example, it is not possible to exclude the effects of pancreas disease, preoperative anticancer agents, or, in some cases, the operative procedure itself from the surgically obtained pancreas samples. Similarly, in studies using autopsied pancreas samples, it is not possible to exclude the effects of nutritional changes or medical treatments prior to death, or the causes of death. Therefore, several approaches are needed to obtain pancreas samples for evaluation of the associations between beta-cell mass and such population characteristics as obesity or diabetes. Table 1 summarizes the studies that have reported associations between the beta-cell mass of human subjects and obesity or diabetes.

### 2.1. Change of Beta Cell Mass in Subjects with Obesity

Previous reports have clearly established that beta-cell mass declines before the onset of T2DM. However, beta-cell mass in subjects with obesity is assumed to increase, since plasma insulin levels in obese subjects increase to compensate for insulin resistance, a process known as hyperinsulinemia [27]. In a rodent study, the beta-cell mass increased threefold in the animals with obesity induced by a high-fat diet [28]. In adult humans, several studies have reported that beta-cell mass increases by approximately 20% to 90% in obese individuals with NDM (non-diabetes) (Table 1), although the definition of obesity differs slightly among these studies [10,11,13,17]. An increase in beta-cell replication was observed in a rodent model of obesity [28], but the source and timing of the increase in beta cells in the obese subjects remain unclear. Since beta-cell replication is more frequently observed in the first five years of life [29,30], childhood obesity or birthweight might affect beta-cell mass. Indeed, we recently reported that the beta-cell area and birthweight were positively correlated in adults with NDM, and the beta-cell area in the group with a history of childhood obesity was increased compared with that in subjects without a history of obesity [26].

### 2.2. Change of Beta Cell Mass in Prediabetes

The next question is how and to what extent beta-cell mass changes during the process of development of glucose intolerance. To answer this question, clinical data of glucose tolerance status are required. Butler et al. have reported that beta-cell mass was decreased by approximately 40% in obese patients with impaired fasting glycemia (IFG) compared with obese NDM [10]. Meier et al. reported a 21% decrease in beta-cell mass in patients with impaired glucose tolerance (IGT) [15]. We recently examined the changes in beta-cell mass according to the glucose tolerance status [25]. The diagnosis of glucose tolerance status in this study was evaluated by the 75-g oral glucose tolerance test before death, and both IFG and IGT were defined as prediabetes. As a result, beta-cell mass decreased with worsening glucose tolerance, from the status of prediabetes. Other reports showed similar trends [12,16,21], and these findings suggest that the stage of glucose intolerance prior to the development of T2DM is also related to reduced beta-cell mass. Based on these findings, a strategy for preventing the development of T2DM must include the maintenance of beta-cell mass even prior to the onset of prediabetes.

### 2.3. Change of Beta Cell Mass in T2DM

Type 1 diabetes (T1DM) is characterized by the loss of insulin secretion due to the destruction of beta cells [31]. Beta cell mass in patients with long-standing T1DM declines to nearly 100% [32,33]. On the other hand, in T2DM it has been considered that defects in insulin action induce hyperglycemia. Since the plasma insulin level of patients with T2DM is often higher than normal (hyperinsulinemia), the beta-cell mass in T2DM is thought to be increased or unchanged. However, several studies have reported that beta-cell mass is decreased by 30–65% in patients with T2DM based on histological analysis (Table 1). In a study using autopsied pancreas samples, Butler et al. reported decreases in beta-cell mass of 41% and 63% in lean and obese patients with T2DM compared with non-diabetic (NDM) controls matched for age and body mass index (BMI), respectively [10]. We have also reported in the surgically resected pancreas that beta-cell mass in patients with T2DM was decreased by 46% compared with that in NDM patients matched for age and BMI [23]. Although the approach for collecting pancreas samples and the ethnic populations differed between these studies, these findings are mostly consistent and suggest that a deficit of beta-cell mass is a common pathophysiological feature of both T2DM and T1DM. In addition, a negative association has been reported between beta-cell mass and duration of diabetes in patients with T2DM [11]. Thus, beta-cell mass might decline with disease progression.

### 2.4. The Mechanism of Beta-Cell Deficit and Change in Alpha-Cell Mass in T2DM

The process of beta-cell deficit in T2DM is assumed to involve a decrease in new beta-cell formation or an increase in beta-cell loss. The replication of other beta cells [34,35] has been proposed as one of the sources of new beta-cell formation. However, we have reported that beta-cell replication in adults is extremely rare [17], and these changes of beta-cell formation in humans remain unclear. In patients with T2DM, beta-cell apoptosis causing beta-cell loss has been shown to be increased [10]. It has been suggested that various factors are involved in beta-cell apoptosis in T2DM, such as hyperglycemia [36], amyloid deposition [37], oxidative [38] or endoplasmic reticulum stress [39], inflammatory cytokines [40], dysfunction of autophagy [41], and lipotoxicity [42]. Several reviews described these molecular mechanisms of beta-cell loss [43,44,45]. As another possible mechanism of beta-cell loss, transdifferentiation of beta cells to alpha cells has been proposed. Although transdifferentiation of beta cells to alpha cells has been observed in rodent studies [46], how often transdifferentiation is occurring in patients with T2DM remains unclear. Cinti et al. have reported that dedifferentiated cells from beta cells increased in diabetics compared with NDM in a study using pancreatic samples from patients with T2DM [47], suggesting that transdifferentiation is the underlying basis of the beta-cell deficit in T2DM. Another group also reported the presence of dedifferentiated beta cells altered by this mechanism in patients with T2DM [48]. However, the contribution of such cells to the deficit of beta cells in T2DM is probably limited, since few dedifferentiated beta cells were reported in this study.

Another controversial question is whether the alpha-cell mass changes in patients with T2DM. Several reports have reported that there is no change in alpha-cell mass and that the ratio of alpha cells to beta cells increases in patients with T2DM [14,23,25]. These findings suggest that beta-cell mass, not the alpha-cell mass, has a major role in the development of T2DM. On the other hand, an increase of alpha-cell mass in T2DM has also been reported [20]. These disparate results regarding the change of alpha-cell mass in T2DM might be partially attributable to the issue of glycemic control. Our recent study found significant positive correlations between alpha-cell mass and glycemic parameters such as HbA1c exclusively in the population with T2DM, not in patients with NGT or prediabetes [25]. In a study reporting an increase in alpha-cell mass in patients with T2DM, the mean HbA1c in the subjects was 7.6% [20], and glycemic control was worse than in our subjects with T2DM [25]. Therefore, greater alpha-cell mass might be associated with poorer glycemic control in patients with T2DM.

### 2.5. Decline in Beta-Cell Function with Worsening Glucose Tolerance

Consistent with the findings on beta-cell mass, a number of studies have reported that beta-cell function is reduced in people with T2DM [49,50,51]. The UK Prospective Diabetes Study showed that beta-cell function assessed by a homeostasis model began to decline prior to disease onset [52]. There are many measures of beta-cell function; the disposition index, for example, is a measure of beta -cell function that is adjusted for insulin sensitivity [53]. DeFronzo et al. reported that the disposition index is decreased in patients with IGT and begins to decline from the stage of NGT [51]. Moreover, a significant positive correlation between beta-cell mass and the disposition index has been reported [25]. These findings indicate that beta-cell function and beta-cell mass are correlated, and both decrease with worsening glucose tolerance. On the other hand, since the functional changes of beta cells have been described as an early predictor of the transition from NGT to IGT [54], the dysfunction of beta cells is suggested to happen at an early stage before the decline of beta-cell mass [55]. As described above, the reduction of beta-cell mass in prediabetes was observed to range from 20% to 40%. However, beta-cell function was shown to be reduced by approximately 50% prior to disease onset [52], suggesting that beta-cell function decreases earlier than beta-cell mass. Furthermore, a rapid recovery of beta-cell function was observed under several conditions. For example, beta-cell dysfunction was improved after overnight beta-cell rest by somatostatin [56]. Taken together, these results indicate that it is difficult to separate beta-cell function and beta-cell mass, although on some occasions they can be dissociated. It has also been reported that beta-cell function declines progressively with disease duration in patients with T2DM [52,57,58,59], which is consistent with the similar progressive decline observed in beta-cell mass [11].

### 2.6. Ethnic Similarities and Differences in the Change of Beta-Cell Mass

The increase in beta-cell mass in obese individuals has mainly been observed in Caucasian populations. Conversely, we reported that there was no increase in beta-cell mass between lean and obese subjects without diabetes in Japanese populations [18,23]. Moreover, these and other Japanese studies have also reported no significant correlation between beta-cell mass and BMI [18,20,21,23]. These results suggest that there are ethnic differences in the change of beta-cell mass in response to obesity. Since the cut-off of obesity in Asian countries is defined as a BMI of 25 kg/m^2^, the lower obesity in Japanese individuals might play a role in these differences of findings in the Caucasian population. However, we showed that beta-cell mass in Japanese individuals without diabetes was not increased by glucocorticoid-induced insulin resistance [22]. Thus, the pathophysiological change of beta-cell mass prior to the development of T2DM might be different between Asians and Caucasians.

A previous meta-analysis reported an ethnic difference in insulin secretion and sensitivity, with Asians exhibiting higher insulin sensitivity and lower insulin secretion compared with Caucasians or Africans [60]. Another study reported that the incidence of T2DM among ethnicities was similar, despite a lesser degree of obesity in the Japanese population [61]. These reports suggest that Asians have less beta-cell regenerative capacity compared with Caucasians, which may induce beta-cell failure and the development of glucose intolerance despite the lower obesity. Although it remains unclear which factors regulate beta-cell mass, there is likely an ethnic difference in the genetic factors involved in this regulation. Numerous susceptibility loci associated with T2DM have been discovered in genome-wide association studies [62], and a general Japanese cohort study showed that the genetic risk score generated using 84 susceptibility loci was associated with the development of T2DM [63]. In addition, most of these loci are suggested to be related to beta cells [64]. Further studies are needed to reveal the association between genetic factors and beta-cell mass.

### 2.7. Change of Pancreas Mass in Obesity and Diabetes

Pancreas mass measured using imaging techniques or autopsy samples has been shown to be increased in subjects with obesity and reduced in subjects with both T1DM and T2DM. Parenchymal pancreas mass in subjects with obesity increased by approximately 10–15%, while pancreatic fat mass increased by approximately 70% in an analysis of computed tomography (CT) images [65]. In a Japanese population, a positive correlation was observed between BMI and pancreas volume measured by CT scan [66], which is in line with the findings in Caucasians. Obesity is a major risk factor for T2DM, and pancreatic fat content increases proportionally with obesity. When the fat supply exceeds the capacity of subcutaneous fat storage, spillover of fat leads to ectopic fat deposits in various tissues, such as the visceral tissues, liver, heart, skeletal muscle, and pancreas [6,67,68,69].

The ectopic fat deposition in various tissues affects tissue dysfunction and metabolic derangements [67], a phenomenon known as the lipotoxicity hypothesis. In a rodent study, excess lipid accumulation in islets of the pancreas was associated with beta-cell dysfunction and apoptosis [6,7]. The elevation in plasma free fatty acid concentration causes insulin resistance [70]. It has also been reported that the incubation of beta cells with free fatty acids impairs insulin secretion and promotes the apoptosis of beta cells [71]. Taylor described vicious cycles between hepatic insulin resistance and beta-cell dysfunction [72]. The hyperinsulinemia caused by obesity increases the conversion of excess calories to liver fat. The fatty liver leads to increased export of VLDL triacylglycerol [73], which increases fat delivery to the pancreatic islets. The excess fatty acid in the pancreatic islets impairs beta-cell function, and the hyperglycemia further increases insulin secretion with consequent enhancement of hepatic lipogenesis. Moreover, leptin, tumor necrosis factor-alpha, and other adipocytokines also secreted from adipocytes within the pancreas may induce beta-cell damage in a paracrine manner [74]. On the other hand, strong correlations between circulating free fatty acids levels and beta-cell dysfunction in humans are lacking [75]. These findings point to a deleterious impact of lipotoxicity due to pancreas fat on beta cells, which needs to be clarified in vivo in more detail. 

Pancreas volume in patients with T1DM has been reported to be decreased by 30–40% [76,77,78], whereas in patients with T2DM it has been reported to be decreased by 8–30% [65,79,80]. Moreover, pancreas volume was reported to correlate negatively with pancreas fat content, and to decrease further with duration of diabetes [80]. Interestingly, a recent study showed an increase of pancreas volume and normalization of pancreas borders by the remission of T2DM with dietary weight loss during 24 months [81]. These studies suggest that the abnormal pancreas morphology is related to the pathophysiology in T2DM, and can be reversed upon remission. In addition, impaired exocrine function has been observed both in patients with T1DM and those with T2DM [82,83]. The pancreas comprises both exocrine and endocrine components. Since animal studies have reported the neogenesis and transdifferentiation in the postnatal period from the exocrine to the endocrine compartment [84,85,86,87,88], the continuous interactions between the endocrine and exocrine pancreas are also suggested in adult humans [74]. Actually, it has been reported in patients with diabetes that pancreas volume is correlated with stimulated C-peptide levels and chymotrypsin activity [89]. To understand the pathophysiology of obesity and diabetes, the mechanisms underlying the interplay between the two compartments of pancreas mass need to be clarified by further investigations.

### 2.8. Association between Pancreas Fat and Glucose Metabolism

The results in humans regarding the relationship between pancreatic fat and glucose metabolism are inconsistent [74]. We have reported that there is no significant difference in the intrapancreatic fat area between subjects with and without diabetes in a histological analysis [90], which is consistent with other histological studies [65,91]. On the other hand, several reports have shown that pancreatic fat increased with the progression of glucose intolerance or T2DM [92,93,94]. Moreover, conflicting results about the association between pancreatic fat content and beta-cell function have also been shown in humans [65,74,95,96,97,98]. One of the reasons for these inconsistencies may be the heterogeneous distribution of pancreas fat. Genetic factors might also play a role. A recent study reported a negative association between pancreatic fat and insulin secretion calculated from a 75-g oral glucose tolerance test in subjects at high genetic risk for diabetes, while subjects with low genetic risk showed a positive correlation [99]. Moreover, Yamazaki et al. recently reported in a population excluding overweight or obese subjects that a higher amount of CT-evaluated pancreatic fat was associated with increased risk of incident T2DM [100]. Further studies using different methods or approaches will be needed to fully elucidate the association between pancreas fat and glucose metabolism.

### 2.9. Beta-Cell Workload Hypothesis

Based on the findings in beta cells, we previously proposed the beta-cell workload hypothesis (Figure 1) and described a treatment strategy for T2DM [101,102]. Since the insulin demand increases in obese subjects, greater insulin secretion from individual beta cells is also required, which named beta-cell workload. It is conceivable that beta-cell function and beta-cell mass decrease through various mechanisms according to beta-cell workload hypothesis [101]. If the beta-cell mass is reduced, the workload becomes greater in each beta cell. In addition, hyperglycemia and high pancreas fat content augment the decline of the decline of beta-cell mass by gluco(lipo)toxicity [36]. This concept suggests that the reduction of beta-cell workload is important for patients with T2DM in order to break the vicious cycle shown in Figure 1.

Figure 2 shows a hypothetical schema of the relationship between beta-cell mass and glucose tolerance in Caucasians and Japanese. Since the demand of insulin increases due to insulin resistance caused by obesity, beta-cell mass increases to adapt to these demands in the Caucasian population. Beta-cell mass is already decreased in the stage of prediabetes, and the workload of each beta cell is assumed to increase chronically. As a result, the overload of beta cells might lead to a reduction in beta-cell mass and beta-cell failure with progression to glucose intolerance. In the Japanese population, because the capacity of beta-cell regeneration is limited compared with that in Caucasians, even the lesser degree of obesity may induce excess beta-cell workload.

The solid bars in Figure 2 represent the change of beta-cell mass with normal glucose tolerance (lean and obese), prediabetes, and T2DM in Caucasians and Japanese when beta-cell mass in the stage of normal glucose tolerance with lean subjects is 100%. NGT: normal glucose tolerance; T2DM: type 2 diabetes.

## 3. Treatment Strategy for T2DM in Relation to the Beta-Cell Workload Hypothesis

Several clinical trials have shown that lifestyle modification and insulin sensitizers, which probably reduce the beta-cell workload, were superior to insulin secretagogues for prevention of the development of T2DM [103,104,105,106,107,108,109]. Lifestyle modification and weight management are the most important factors in the treatment for T2DM at any stage, including the diseases pre-onset stage. It has been reported that an intensive lifestyle intervention increased the rate of remission of T2DM compared with diabetes support and education [110]. The Diabetes Remission Clinical Trial showed that nearly half of patients diagnosed with T2DM within the six years prior to the study could be returned to long-term non-diabetic glucose control by using intensive weight management in the context of routine primary care [111]. Moreover, this trial demonstrated that the ability to recover first-phase insulin was increased in responders who returned to non-diabetic glucose control after weight loss [112]. These findings suggest that lifestyle intervention with weight loss at an early stage of diagnosis may prevent ongoing loss of beta-cell mass and function. Since metformin reduces beta-cell workload by suppressing insulin demand through a reduction of hepatic glucose production, an early start to metformin therapy should be considered. Incretin therapy enhances insulin secretion in a glucose-dependent manner and reduces glucagon secretion [113]. The use of dipeptidyl peptidase-4 (DPP-4) inhibitors has been shown to achieve better glycemic durability compared with treatment using sulfonylureas [114,115]. In addition, glucagon-like peptide-1 (GLP-1) receptor agonists induce weight loss by slowing gastric emptying and suppressing appetite. Sodium-glucose cotransporter 2 (SGLT2) inhibitors also reduce body weight by increasing glucose excretion in urine. It has been shown that cardiovascular outcomes were improved by treatment using SGLT2 inhibitors and GLP-1 receptor agonists in clinical trials [116,117,118,119]. The American Diabetes Association recommends the priority use of these drugs for the treatment of T2DM patients with atherosclerotic cardiovascular disease, chronic kidney disease, or heart failure [120]. In a recent updated meta-analysis of 30 randomized cardiovascular outcome trials, treatment strategies that lower bodyweight, including therapy with intensive lifestyle modification, SGLT2 inhibitors, and GLP-1 receptor agonists, have been shown to reduce the risk for fatal and non-fatal atherosclerotic events and heart failure [121]. The consideration of the use of these drugs, which result in the reduction of beta-cell workload through the loss of weight, is consistent with our proposed treatment strategy. Since it has been shown that insulin therapy maintains beta-cell function [122], insulin therapy might reduce the workload of beta cells despite the risk of weight gain. In most cases, combination therapy should be considered. In a recent clinical trial, early combination therapy with vildagliptin and metformin showed superior glycemic durability compared with the initial metformin monotherapy for patients with newly diagnosed T2DM [123]. Thus, early active intervention for patients with T2DM is required before the beta-cell workload becomes excessive.

## 4. Conclusions

This review summarized the current knowledge on beta-cell mass, beta-cell function, and pancreas fat in obesity or T2DM, and the treatment strategy for T2DM in relation to beta cells was discussed. Since Asians seem to have less beta-cell functional capacity compared with Caucasians, a therapeutic strategy for T2DM based on the beta-cell workload hypothesis should be emphasized for Asians. An early treatment that includes lifestyle modifications to preserve beta-cell mass or beta-cell function is needed in order to counter the pandemic burden of T2DM.

## Figures and Tables

**Figure 1 nutrients-12-03846-f001:**
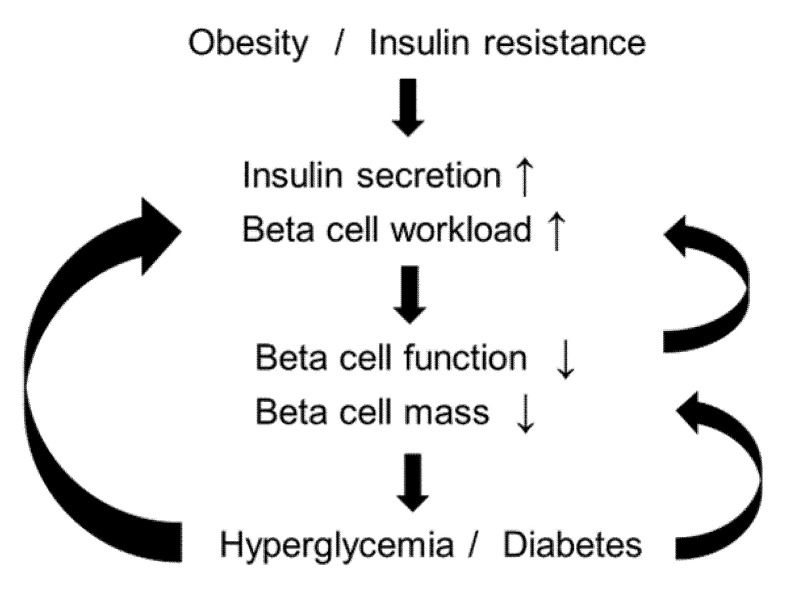
Proposed schema of the development of type 2 diabetes in relation to beta-cell workload.

**Figure 2 nutrients-12-03846-f002:**
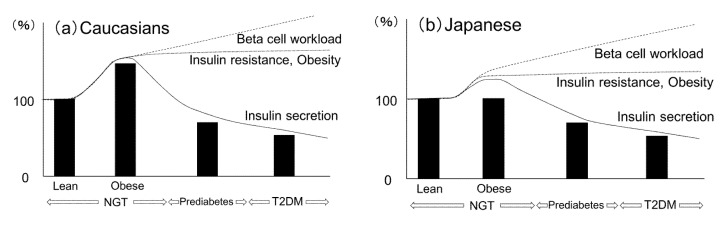
Hypothetical schema of changes in beta-cell mass during obesity and the development of glucose intolerance in (**a**) Caucasians and (**b**) Japanese subjects.

**Table 1 nutrients-12-03846-t001:** Studies on the associations between the beta-cell mass of human subjects and obesity or diabetes since 2000.

Study Year, Reference	Approach for Sample Collection	Country Where Performed (Ethnicity)	Metabolic Status/ Number of Samples (Male)	Mean Age (Year)	Mean BMI (kg/m^2^)	Change in Beta-Cell Mass (BCM) or Beta-Cell Area (BCA)	
						Obesity	Diabetes
Sakuraba et al., 2002, [8]	autopsy	Japan (n.a.)	NDM 15 (10)	52	21.3	n.a.	BCM: 1.14 g
			DM 14 (9)	61	20.7		BCM: 0.82 g * 30% decreased vs. NDM
Yoon et al., 2003, [9]	organ donor	Korea (n.a.)	9 (6)	41	23.8	Positive correlation between BCA and BMI	n.a.
	pancreas surgery		NDM 10 (6)	57	22.2	No difference	BCA: 1.94%
			T2DM 25 (15)	60	22.2	Positive correlation between BCA and BMI	BCA: 1.37% * 29% decreased vs. NDM
Butler et al., 2003, [10]	autopsy	USA (n.a.)	NDM lean 17 (10), BMI < 25 kg/m^2^	78	<25 kg/m^2^	BCA:1.71%	
			T2DM lean 16 (9), BMI < 25 kg/m^2^	80		n.a.	41% decreased vs. NDM Lean
			NDM Obese 31 (16), BMI > 27 kg/m^2^	67	>27 kg/m^2^	BCA:2.6%	
			IFG Obese 19 (10), BMI > 27 kg/m^2^	63		n.a.	BCA: 1.56% * 40% decreased vs. NDM Obese
			T2DM Obese 41 (24), BMI > 27 kg/m^2^	63			BCA: 0.96% * 63% decreased vs. NDM Obese
Rahier et al., 2008, [11]	autopsy	Belgium (European)	BMI < 25 kg/m^2^, NDM 26 (14)	63	21.9	20% increased in NDM	41% decreased vs. NDM with BMI < 25 kg/m^2^
			BMI < 25 kg/m^2^, T2DM 15 (5)	72	22.3		
			BMI 26–40 kg/m^2^, NDM 25 (21)	68	29.9		38% decreased vs. NDM with BMI 26-40 kg/m^2^
			BMI 26–40 kg/m^2^, T2DM 34 (21)	68	31.7		
Meier et al., 2009, [12]	pancreas surgery	Germany (n.a.)	NGT 8 (3)	56	23.6	n.a.	BCA: 1.22%
			IGT/IFG 14 (6)	63	23.4		BCA: 1.14%
			DM 11 (8)	56	24.1		BCA: 0.43% * 65% decreased vs. NGT
Hanley et al., 2010, [13]	organ donor	Canada (n.a)	NDM Lean 18 (13), BMI < 27 kg/m^2^	59	24.2	BCA:1.15%	
			T2DM Lean 8 (6), BMI < 27 kg/m^2^	59	23.7	n.a.	BCA: 1.28%
			NDM Obese 23 (10), BMI > 27 kg/m^2^	59	31.4	BCA: 2.20% * 91% increased vs. NDM lean	BCA: 2.20%
			T2DM Obese 11 (5), BMI > 27 kg/m^2^	61	32.6	n.a.	BCA: 1.41% * 36% decreased vs. NDM Obese
Henquin et al., 2011, [14]	autopsy	Belgium (n.a.)	NDM 52 (35)	66	25.8	No difference	36% decreased vs. NDM
			T2DM 50 (26)	68	30.1	No difference	
Meier et al., 2012, [15]	pancreas surgery	Germany (n.a.)	82 (42)	60	24.4	n.a.	BCA in IGT 21% decreased vs. NGT
Yoneda et al., 2013, [16]	pancreas surgery	Japan (n.a.)	NGT 11 (7)	67	21.1	n.a.	BCA: 1.60%
			IGT 11 (3)	67	22.7		BCA: 0.99% * 38% decreased vs. NGT
			Newly Diagnosed DM 10 (4)	66	23.4		BCA: 0.93% * 42% decreased vs. NGT
			Long-Standing T2DM 10 (6)	76	20.5		BCA: 0.53% * 67% decreased vs. NGT
Saisho et al., 2013, [17]	autopsy	USA (n.a.)	NDM Lean 53 (30), BMI < 25 kg/m^2^	37	21.2	BCM: 0.8 g	n.a.
			NDM Obese 61 (43), BMI ≥ 27 kg/m^2^	41	35.1	BCM: 1.2 g* 50% increased vs. Lean	
Kou et al., 2013, [18]	autopsy	Japan (Japanese)	NDM Lean 39 (22), BMI < 25 kg/m^2^	47	20.4	BCM: 0.7 g	n.a.
			NDM Obese 33 (24), BMI ≥ 25 kg/m^2^	47	28.5	BCM: 0.6 g No difference vs. Lean	
Mezza et al., 2014, [19]	pancreas surgery	Italy (n.a.)	NDM 18 (9)	53	27.9	BCA in insulin-resistant ^#^ (1.10%) increased vs. insulin-sensitive ^#^ (0.58%*)	n.a.
Mizukami et al., 2014, [20]	autopsy	Japan (n.a.)	NDM 30 (21)	65	22.4	No difference	BCM: 1.86 g
			DM 47 (38)	68	22.7		BCM: 1.27 g * 32% decreased vs. NDM
Fujita et al., 2015, [21]	pancreas surgery	Japan (n.a.)	NGT 13 (8)	64	21.5	No difference	BCA: 1.072%
			IGT 9 (4)	61	20.8		BCA: 0.998%
			DM 10 (7)	68	22.4		BCA: 0.762%
Sato et al., 2015, [22]	autopsy	Japan (Japanese)	NDM 26 (15)	63	20.8	n.a.	BCA:1.66%
			DM 25 (21)	66	21.5		BCA: 0.92% * 45% decreased vs. NDM
Inaishi et al., 2016, [23]	pancreas surgery	Japan (Japanese)	NDM Lean 40 (17), BMI < 25 kg/m^2^	64	21.5	BCA:1.42%	BCA: 1.48%
			NDM Obese 10 (9), BMI ≥ 25 kg/m^2^	63	26.4	BCA: 1.71% No difference vs. Lean	
			DM 49 (35)	67	21.9	n.a.	BCA: 0.80% * 46% decreased vs. NDM
Xin et al., 2017, [24]	autopsy	Japan (n.a.)	NDM 22 (11)	61	21.8	n.a.	BCA in T2DM 30% decreased vs. NDM
			T2DM 27 (19)	63	22.8		
Inaishi et al., 2020, [25]	autopsy	Japan (Japanese)	NGT 40 (24)	80	20.4	No difference	BCA: 1.85%
			Prediabetes 31 (25)	78	22.1		BCA: 1.59%
			T2DM 32 (19)	76	23.6		BCA: 1.17% * 37% decreased vs. NGT
Sasaki et al., 2020, [26]	pancreas surgery	Japan (Japanese)	NDM 38 (20)	61	22.3	No difference	BCA: 1.14%
			DM 26 (23)	67	25.1		BCA: 0.75% * 34% decreased vs. NDM

* Significant at *p* < 0.05. ^#^ Insulin sensitivity or resistance measured by the euglycemic hyperinsulinemic clamp procedure. n.a.: not available; BMI: body mass index; BCM: beta-cell mass; BCA: beta-cell area; NDM: non-diabetes; DM: diabetes mellitus; T2DM: type 2 diabetes; NGT: normal glucose tolerance; IFG: impaired fasting glycemia; IGT: impaired glucose tolerance.

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
