# Peer review of "Beta-Cell Mass in Obesity and Type 2 Diabetes, and Its Relation to Pancreas Fat: A Mini-Review"

_nutrients, 2020, doi:10.3390/nu12123846_

Round 1

Reviewer 1 Report

Summary: In this review the authors conducted a Meta-analysis on previous studies on the role of beta-cell mass and its contribution to the development of obesity-associated Type 2 diabetes Mellitus. In particular the author discuss the importance and mechanisms behind beta-cell failure in the development of the obesity-associated type 2 diabetes mellitus (T2DM).

The paper is well-written explaining a significant climax for the role of beta-cell mass in the development of T2DM. The manuscript also touches up on the decline of function, but the authors predominantly focus their review of the studies on be-cell mass. I have some minor points for the authors on background information and explanation to improve the manuscript. Concerning the background information and knowledge, I have some comments on the information that should be added to the text.

Major points:

The authors should work on better organizing the different sections in this manuscript so that the information they are providing flows. And that the timing of the different stages of beta-cell mass evolution in the development of T2DM is clear for the reader: Obesity… beta cell compensation…decline in beta-cell function…beta-cell mass reduction…development of hyperglycemia and development of diabetes. Although this is over-simplified it’s just to give the authors an idea of the type of flow I am talking about. This is a suggestion, the authors can decide about the better way to present their thinking in relation to the literature.

Perhaps section “2.2. Change of beta cell mass in prediabetes” should be moved before section 2.1. Change of beta cell mass in T2DM.

Page 9: Section 2.4 needs to be expanded. The authors should provide a better discussion of the relationship between beta-cell function and mass, and mention the phenomena of beta-cell exhaustion and its possible role in beta-cell function decline.

Line 120-122: “These findings indicate that beta cell function and beta cell mass are correlated, and both decrease with worsening glucose tolerance. It has also been reported that beta cell function declines progressively with disease duration in patients with T2DM [44,46–48].

There is a consensus that beta-cell function starts to decline way before the IGT sets in. I agree that beta-cell function and mass are correlated in T2DM, however, the timing of when each one starts is believed to be different. It is suggested that the decline of beta-cell function and exhaustion happens at a very early stage before beta-cell loss. Perhaps the authors should expand more on this.

Here are some references that might help:

1 Chunguang Chen et al., Molecular Metabolism, 2017, 6(9): 943-957

2 Gordon C. Weir and Susan Bonner-Weir, Diabetes 2004, 53(suppl 3): S16-S21

Again section “2.5. Change of beta cell mass in subjects with obesity” should be part of the prediabetes stage.

Section “3. The role of fat deposition in the pancreas” should be before the section “2.6. Ethnic similarities and differences in the change of beta cell mass”.

Minor points.

Page 1: Line 24: …or the cost of treatment. Change this sentence to …and increased health care cost,…

Page 8: Lines 77-78: …was evaluated by the 75-g oral glucose tolerance test before death,…

            Line 95: …[38], how often transdifferentiation is occurring…

            Line 99-100: “On the other hand, another group reported that only a small number of endocrine cells are altered by this mechanism [40], suggesting that the contributions for the deficit of beta cells in T2DM are limited.”

This sentence is somewhat confusing. Are the authors talking about the mechanism of dedifferentiation of beta cells into other cells, having a limited contribution to the beta-cell deficit seen in T2DM? Or other endocrine cells and their contribution to beta-cell deficits. The meaning of this sentence is not clear. The authors need to rewrite this sentence with some more clarity.

Section “3.1. Change of pancreas mass in obesity and diabetes” only provide information on the changes in pancreas mass in obesity and diabetes. How is that important for fat deposition in beta-cells? The authors should discuss this further and add more references on this.

Page 9: Line 185: Pancreas mass in patients with T1DM has been reported to be decreased by 30%–40% [60-62], whereas in patients with T2DM it has been reported to be decreased by 8%–20% [58,63].

Reviewer 2 Report

The review entitled "Beta cell mass in obesity and type 2 diabetes, and its relation to pancreas fat: A mini-review" (ID-Nutrients 1021493) is quite interesting, written in good English, it has examined numerous articles on the subject which took into account data from both autopsy and pancreas surgery and organ donors. Nonetheless, in my opinion, the reader acquires information that has long been known and widely published in numerous other articles. On page 1. Introduction - line 23-24 - I don't find very clear the sentence “Since diabetes mellitus is associated with…. [2-4] or the cost of treatment, ....  In section 2.2. Change of beta cell mass in prediabetes - line 78 - is not clear to me: .... before death, and both IFG and IGT ......  The journals cited in the References section should be reordered with the right international abbreviations or with the capital letters. 

Round 2

Reviewer 2 Report

The article in question has been improved in style and content and has been made more explanatory for the reader. I think the changes made by the authors are satisfactory.
